



# Ice thickness of Comox and Kokanee Glaciers, British Columbia, determined through relative gravity surveys

Matthew R. G. Forbes[1] and Lucinda J. Leonard[2]

[1]Department of Physics & Astronomy, University of Victoria, Victoria, British Columbia V8P 5C2, Canada
[2]School of Earth and Ocean Sciences, University of Victoria, Victoria, British Columbia, V8P 5C2, Canada

**Correspondence:** Matthew R. G. Forbes (mattforbes@uvic.ca)

**Abstract.** Ice thickness data are sparse or lacking for many glaciers worldwide, making it difficult to track changes in ice volume due to ongoing climate change. In this study, we collect and model relative gravity survey data, to provide the first estimate of ice thickness for the retreating Comox Glacier, a historically important source of freshwater for eastern Vancouver Island, British Columbia. We validate our approach by carrying out a similar analysis across the Kokanee Glacier, for comparison with recent ice penetrating radar results. Modelling of the Bouguer gravity anomaly across each glacier provides an average inferred ice thickness of $42 \pm 4$ m across a 450 m transect of Comox Glacier, and $50 \pm 3$ m across a 220 m transect of Kokanee Glacier, consistent with previous measurements. Future repeat surveys will enable monitoring of ice thickness changes over time. Compared to other methods, gravity surveying offers a lower cost and logistically simpler alternative for the collection of ice thickness data on glaciers worldwide.

## 1 Introduction

Ongoing reduction in permanent ice volume has been consistently witnessed in western North America (Bevington and Menounos, 2022; Menounos et al., 2025) and across most glacial systems around the world (e.g., Taillant, 2021). Beyond the environmental impacts of such changes, billions of people worldwide are currently dependent, at least in part, on glacial systems for sufficient water supply, agricultural output, and economic stability (Scott et al., 2019). For 45,000 residents on eastern Vancouver Island in British Columbia, Canada, the water supply has historically been buffered by the now receding Comox Glacier (CG; Fig. 1) (Comox Valley Regional District, 2022; Kloster, 2021). Although no ice thickness measurements of CG have been made to date, a high-resolution regional glaciation model constrained by satellite imagery of both the Coast Range and Vancouver Island mountains projected an approximately 50% loss of ice coverage by 2050 (Clarke et al., 2015). While the loss of the glacier may not be immediately disastrous for the regional population since much of the water supply is fed through precipitation on the surrounding watershed, it will put more pressure on a vital resource for an increasingly populated region as climate change continues and summer dry periods in the area lengthen (BC Agriculture & Food Climate Action Initiative, 2020).

On a broader scale beyond the Comox region, of the ~200,000 glaciers globally, measurements of ice thickness have been compiled for only around 3000 (GlaThiDa Consortium, 2020). The global ice thickness database is composed predominantly



of airborne and terrestrial ice penetrating radar measurements (5061 surveys: 98.4%), with the remainder of contributions from seismic reflection (43 surveys), electromagnetic sounding (2), and drilling (18), the latter being the only truly direct method (GlaThiDa Consortium, 2020; Welty et al., 2020). These various methods, although accurate and relatively simple, are often expensive, time consuming, and/or require bulky equipment that may be difficult to transport. Ice thickness can also be estimated indirectly, with methods based on glacier surface area (via area-volume scaling) or surface slope (e.g., Grinsted,

2013; Huss and Farinotti, 2012). However, such approaches tend to overestimate ice thickness by ~20-30% compared with in situ observations (Gärtner-Roer et al., 2014). Uncertainties are reduced with the use of a multiple-model ensemble, but in situ data remain critical for validation (Farinotti et al., 2019). To better supply communities with knowledge of their local water system, as well as provide more data to constrain models of glacier and climate dynamics, there is a need for additional methods to more directly infer glacial ice thickness.

Gravity surveying was first used to determine the ice thickness of a valley glacier by Bull and Hardy (1956); it remained a popular method through the 1960s and 1970s (e.g., Russell et al., 1960; Kanasewich, 1963; Corbató, 1965; Hyndman, 1965; Klingele and Kahle, 1977), but has been rarely used since then (e.g., Borthwick et al., 2025). Past studies found generally good agreement between gravity-derived ice thickness and that determined from seismic surveys and/or boreholes (Kanasewich, 1963; Corbató, 1965; Klingele and Kahle, 1977; Caldwell, 2005). A drawback of the gravity method in the past was the need

for tedious and time-consuming terrain corrections, involving manual assessment of elevations at radial distances from all measurement points, from paper topography maps (e.g., Kanasewich, 1963) – such corrections are now much simpler and quicker, using digital elevation models and standard computer routines (e.g., Cella, 2015). Gravity equipment is more portable than other in situ instruments, and surveys can be completed relatively quickly with a team of just two people, suggesting that gravity surveying is an overlooked but valuable method of determining glacial ice thickness.

The main objective of this study is to provide a first baseline measurement of ice thickness for Comox Glacier, using the relatively inexpensive method of ground-based relative gravity surveying. In addition to collecting and modelling a gravity transect across Comox Glacier, we also collect a similar profile across Kokanee Glacier (KG; Fig. 1), where the gravity-derived ice thickness can be compared with thickness values recently determined using ice penetrating radar (Pelto et al., 2020).

## 2 Glacier setting

Comox Glacier (49.550°N, 125.355°W) is located within the Vancouver Island Ranges near the south-eastern border of Strathcona Provincial Park between Black Cat Mountain and Mount Arthur Evans. The bedrock of the region is primarily basalt of the middle to upper Triassic Karmutsen Formation (Vancouver Group) (Greene et al., 2010; Massey et al., 2005). The median, minimum, and maximum elevations of the glacier were estimated through the Randolph Glacier Inventory 6.0 (RGI 6.0) at 1840 m, 1490 m, and 1940 m, respectively, over an area of approximately 1.55 $km^2$ (RGI Consortium, 2017). An average

elevation was measured for the survey area at 1872 $\pm$ 2 m via a handheld Garmin ETREX 32x Global Positioning System (GPS) device.



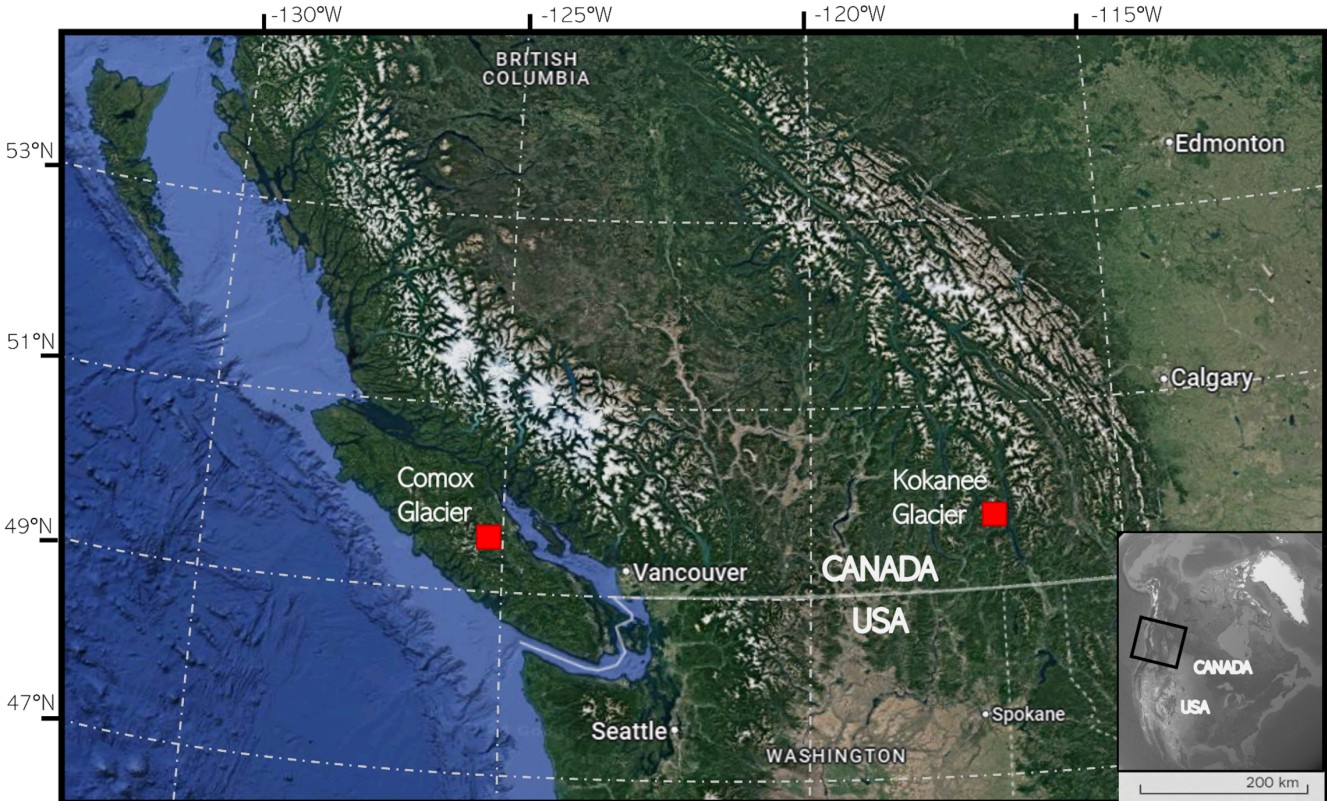

**Figure 1.** Location of Comox Glacier and Kokanee Glacier (red squares) within British Columbia, Canada. Basemaps prouced on © Google Earth, with satellite imagery from Landsat/Copernicus up to 01/01/2021.

Within mainland BC, Kokanee Glacier (49.749°N, 117.145°W) sits inside Kokanee Glacier National Park in the West Kootenays Selkirk mountain range on the north face of Kokanee Peak. Kokanee Peak and the surrounding mountains are composed of middle Jurassic granite and granodiorite of the Nelson batholith (Brown and Logan, 1989; Massey et al., 2005; Logan et al., 1988). Occurrences of Triassic argillite, quartzite, and limestone, associated with the Slocan group to the northeast of the batholith, are also dispersed throughout the park (Logan et al., 1988; Brown and Logan, 1989). The median, minimum, and maximum elevations of the glacier were again estimated through the RGI 6.0 dataset at 2620 m, 2230 m, and 2800 m over an area of 2.71 $km^2$ (RGI Consortium, 2017). The average survey area elevation was recorded at 2659 ± 2 m.





## 3  Methods

### 3.1  Gravity field surveys

Both the CG and KG survey areas were accessed on foot, over a 2-day return trip from Comox Glacier and Gibson Lake trailheads, respectively. In both cases the survey was completed in the late dry season prior to any snowfall. Conditions on both occasions were well above freezing such that ice was witnessed to melt while the survey was taking place. Surface runoff was observed draining from the head of KG, and within both glaciers running water could be heard below the surface. Many small crevasses were also observed along the transect area, with lengths and depths of at least a few meters.

To take the relative gravity measurements, we used a LaCoste & Romberg Model G gravity meter. This instrument, accurate to $\sim 0.01$ mGal, measures relative gravitational acceleration via the mechanical response of a mass on a beam supported by a "zero-length" spring which is clamped during transportation, making it sufficiently rugged and portable for on-foot traversal of mountain environments without damage (LaCoste & Romberg, 1989).

### 3.1.1  Comox Glacier

The gravity survey for CG was carried out on the afternoon of 10 October 2022. The survey transect extended from a point located 2 m south of the glacier edge to the approximate center of the glacier. A reference location was set up on the exposed rock next to CG, shown as location "1, 12" of Fig. 2a. The first and last gravity measurements were made at this base station to allow for temporal corrections to be calculated and applied to the survey data (see Sect. 3.2).

During a two hour period, a total of twelve gravity readings were recorded over 450 m traversing the straightest path which could be safely crossed from the base station at the edge extending towards the center of the glacier. For each measurement the gravity meter was placed on a metal plate dug a few centimeters into the ice. In addition to the gravity readings, the time, GPS location, and elevation of each measurement were recorded (see Supplement Table S1). The relative horizontal and vertical positions of the measurement sites were then surveyed using a Spectra Precision Focus 6 5 in. Total Station and associated reflector prism.

To ensure consistency and that the apparatus was undamaged for the experiment, measurements were also made at a location in Victoria, BC, 2 days before and 1 day after the survey (see Table S1, S2 and Sect. S2.1.1 of Supplement).

### 3.1.2  Kokanee Glacier

KG was accessed on the afternoon of 1 September 2023. The transect area was chosen of an approximately flat 220 m section on the upper part of the glacier, again extending from a base station on exposed rock approximately 3 m west of the glacier edge. The process of the experiment was completed nearly identically to that of Comox Glacier, except that a total of 11 measurements were made (see Supplement Table S1), and the Total Station was exchanged for a more easily portable Bosch handheld 400 ft. laser rangefinder for measurements of relative height and distance along the transect. Again, the first and last





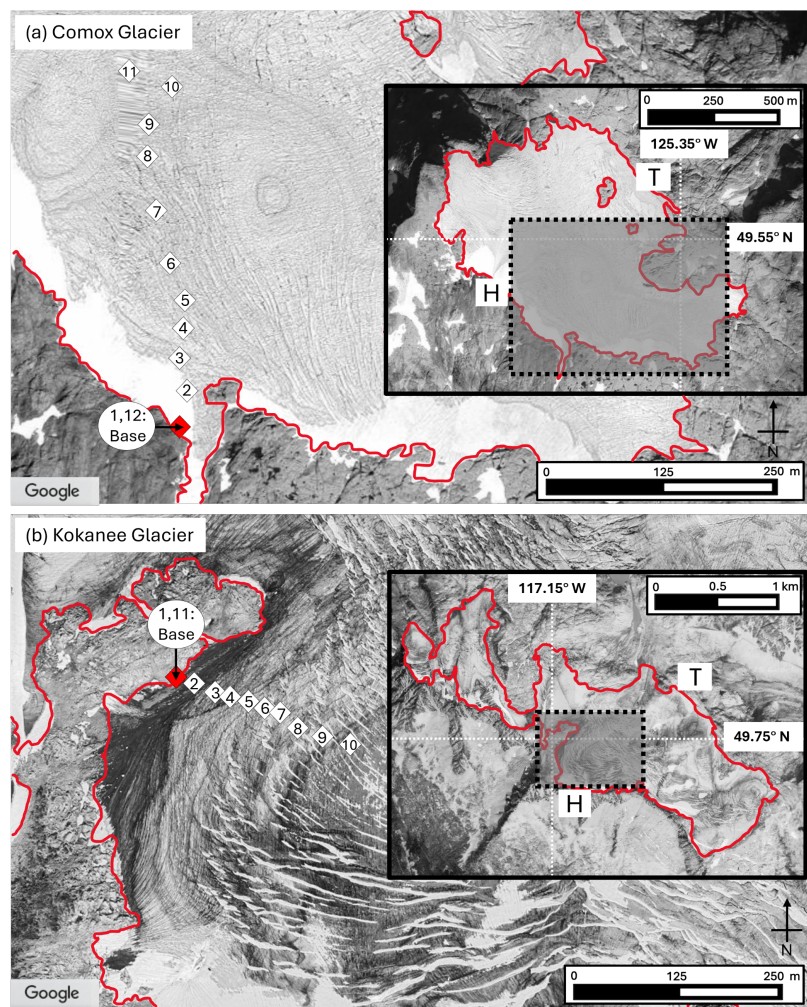

**Figure 2.** Survey measurement locations (diamonds) across Comox Glacier (a) (© Google Earth, Airbus, 30/09/2023) and Kokanee Glacier (b)(© Google Earth, Airbus, 08/09/2024). For both (a) and (b) the location of the larger image is given by the dashed line box in the inset map, which includes the full glacier extent, with the head and toe regions marked by H and T respectively.

measurements, labelled "1, 11" in Fig. 2b, were made at the base station, and for consistency extra measurements were made before and after the survey for comparison.

## 3.2 Data reduction & density modelling

To isolate the local Bouguer anomaly resulting from the negative density anomaly presented by each glacier, and thus enable the approximation of ice thickness via modelling, standard corrections were made to both sets of gravity data (see Sect. S1 and



S2 in the Supplement). Corrections included those for the effects of solid Earth tides, linear drift from gravity meter spring

creep, latitude, elevation (free air correction), the Bouguer plate correction, and the local terrain (see Table S2 in Supplement).

Firstly, manufacturer instructions specific to the LaCoste & Romberg Model G instrument were followed to convert gravity readings to units of mGal (LaCoste & Romberg, 1989). Tidal corrections were then calculated through a MATLAB program adapted from code produced by Ahern (1993) with equations based on Schureman (1924) and compiled by Longman (1959).

For both the terrain and Bouguer plate corrections a rock density of $2.89\pm0.16$ g cm$^{-3}$, associated with the predominant

basalt of the Karmutsen Formation surrounding CG, and $2.67\pm0.08$ g cm$^{-3}$, associated with the predominant granite and granodiorite makeup of the Nelson batholith surrounding KG, were used (Enkin, 2018; Sharma, 1997). These density values are the average saturated bulk density of, respectively, Karmutsen basalt samples and British Columbia granite and granodiorite samples, as documented in the Canadian Rock Physical Property Database (Enkin, 2018). For both glacier surveys the terrain corrections were calculated using GTeC software (Cella, 2015) alongside the Province of British Columbia Terrain Resource

Information Management 1:20,000 topographic dataset at a pixel size of 2 m and elevation grid of 25 m (GeoBC, 2013). A radius of 7 km was used around each measurement station, with gravity corrections compiled over the area with step size increasing from 25 m to 2 km with distance from each station. See Sect. S1 in the Supplement for the full list of measurements and corrected data, and Sect. S2 for the associated equations.

Final determination of the ice thickness across the CG and KG transects was inferred through gravity modelling with Grav-

Mag software (Table 1, Fig. 3) (Burger et al., 2006). The software calculates the gravity contribution of a 3-D ice body of given dimensions (represented in 2-D as adjoining polygons) relative to underlying and adjacent rock, for comparison with the observed Bouguer gravity anomaly. The assumed homogeneous ice density is 0.917 g cm$^{-3}$, and rock densities are as defined above for the CG and KG study areas, such that the ice body is assigned a density contrast of $-1.97\pm0.16$ g cm$^{-3}$ and $-1.75$ $\pm0.08$ g cm$^{-3}$ for CG and KG, respectively (Enkin, 2018; Sharma, 1997).

The glacier width parallel to the survey transect is constrained by the satellite imagery and the modelling assumes uniform continuation of the ice body in the perpendicular directions. In each case ice body thickness was then varied along the profile in order to minimize the root mean square error (RMSE) misfit between the observed and model-predicted values of relative gravity. Once the RMSE misfit was minimized for each model, the inferred ice thickness at each measurement location was recorded (Table 1). Uncertainties in ice thickness were determined by finding the minimum and maximum modelled values

that provided a best fit to each gravity data point plus and minus its uncertainty.

## 4 Results

The resultant local Bouguer gravity anomaly values along each transect are listed in Table 1. Across the CG, relative gravity decreases with distance from the base station, with a gradual decrease to approximately -1.0 mGal at 130 m, and a steeper decline over the next 70 m to reach an average of $-2.95 \pm 0.18$ mGal for the remainder of the profile (Fig. 3a). Forward

modelling with the GravMag software shows a best-fit average ice thickness of less than 10 m over the first 130 m, where the inferred ice thickness then increases to an average of $42 \pm 4$ m, with an RMS misfit of 0.064 mGal. A maximum ice thickness





**Figure 3.** GravMag density structure model of ice thickness across Comox Glacier (a) and Kokanee Glacier (b), providing a best fit to the measured Bouguer gravity anomaly data (Burger et al., 2006). The top panel of each sub-figure is the observed gravity (triangles) versus distance along the transect, plotted alongside the calculated gravity (stars) produced by the structure in the lower panel. For the model in the bottom panel, the white region indicates rock, while the grey bodies/polygons indicate ice. On the left hand side of the figure the ice has been modelled to mimic the glacier edge, while on the right hand side the full known lateral extent of the ice body is not shown, but is included in the model.

of $57 \pm 2$ m is inferred just over 200 m from the base station, 250 m south of the center point of the glacier (point 6-7, Fig. 2a).

Completing a similar analysis for KG, the Bouguer gravity profile shows a steady decrease with distance across the glacier from the base station. The profile levels out after approximately 50 m to a low of -3.19 $\pm$ 0.13 mGal relative to the base station (Fig. 3b). The best-fit forward model infers an ice thickness range and average for the central region of the transect of 42-56 $\pm$ 2 m and $50 \pm 3$ m, respectively, at an RMS misfit of 0.24 mGal (points 3-10, Fig. 2b).



**Table 1.** Bouguer gravity anomaly values ($\delta g^{boug}$), inferred glacier ice thickness ($Tck.$), and distance from base station ($Dst.$) for Comox Glacier (CG) and Kokanee Glacier (KG). See Fig. 2 for measurement locations.

| | 1 | 2 | 3 | 4 | 5 | 6 | 7 | 8 | 9 | 10 | 11 | 12 |
|---|---|---|---|---|---|---|---|---|---|---|---|---|
| $\delta g^{boug,CG}$ $\pm 0.15$(mGal) | 0.00 | -0.62 | -0.86 | -1.01 | -2.44 | -3.40 | -2.97 | -2.68 | -3.13 | -3.05 | -3.02 | 0.00 |
| CG $Tck.$ $\pm 2$(m) | 0 | 4 | 7 | 3 | 35 | 57 | 36 | 30 | 46 | 45 | 45 | 0 |
| CG $Dst.$ $\pm 0.1$(m) | 0.0 | 37.4 | 84.5 | 120.2 | 155.9 | 201.5 | 274.0 | 337.4 | 409.2 | 438.3 | 445.9 | 0.0 |
| $\delta g^{boug,KG}$ $\pm 0.12$(mGal) | 0.00 | -2.18 | -2.71 | -2.90 | -3.22 | -3.34 | -3.20 | -3.47 | -3.42 | -3.29 | 0.00 | - |
| KG $Tck.$ $\pm 2$(m) | 0 | 31 | 42 | 44 | 51 | 55 | 50 | 53 | 56 | 51 | 0 | - |
| KG $Dst.$ $\pm 0.1$(m) | 0.0 | 25.9 | 52.3 | 70.4 | 88.8 | 110.2 | 129.4 | 159.4 | 187.6 | 224.1 | 0.0 | - |

## 5 Discussion

### 5.1 Study limitations

As discussed in sect. 5.2, the ice thickness determined for Kokanee Glacier is similar, though slightly lower, than previous radar-based measurements. It is possible that our models under- or over-estimate the true thickness of Kokanee and Comox Glaciers, due to necessary simplifying assumptions. The modelled value for the glacier density assumes a homogeneous ice body, when in fact the glacier contains some volume of sediment and meltwater (both denser than ice) as well as air-filled crevasses (less dense than ice). We assume that these volumes are relatively insignificant and that they mostly cancel each 145 other out. Adding such features to the model may enable a different ice thickness profile that equally well fits the data, but the inclusion of unconstrained complexities is not warranted. Additional sources of uncertainty include the assumption of homogeneity of the adjacent rock density, as well as the choice of total area and grid size for the terrain corrections.

Our study is somewhat limited in that only one transect was surveyed for each glacier. Thus, the mean ice thickness values for each profile may not be representative of the mean ice thickness of the glacier overall - a common issue for in situ glacier 150 measurements, which tend to be biased to the more accessible parts of the glacier, avoiding dangerously steep, crevassed, or debris-covered slopes (e.g., Gärtner-Roer et al., 2014). However, since each transect crosses the main trunk of the glacier in the midstream section, the measured gravity anomaly results partially from the density structure in the adjacent up- and downstream directions, as well as directly along the transect. Modelling then enables approximation of an ice thickness profile that is spatially averaged in the up- and downstream directions. Gravity surveying of more complex glaciers would benefit from 155 modelling in three dimensions, as in the study of Borthwick et al. (2025).

### 5.2 Glacier ice thickness comparison

For Comox Glacier, we provide the first ever determination of ice thickness that can be used in regional or global modelling efforts (e.g., Farinotti et al., 2017) and to serve as a baseline value to enable monitoring of future change; however, there are no previous estimates for comparison. The CG ice thickness values (mean $42 \pm 4$ m; maximum $57 \pm 2$ m) are consistent with





those measured in glaciers of comparable area (average mean thickness of $40 \pm 17$ m for 101 1-2 km$^2$ glaciers; GlaThiDa Consortium, 2020).

Kokanee Glacier was extensively surveyed using ice penetrating radar in 2017 (Pelto et al., 2020). That study measured ice thickness up to 88 m, with an average of 48 m, consistent with the $50 \pm 3$ m average determined for our gravity transect, and with the average mean thickness of $50 \pm 20$ m for 66 2-3 km$^2$ glaciers within the GlaThiDa dataset (GlaThiDa Consortium, 2020). Our estimated uncertainties at 6-10% are also similar to those estimated for the radar-derived ice thicknesses (5.2-10.4%; Pelto et al., 2020) and for gravity studies of other glaciers (e.g., Kanasewich, 1963; Corbató, 1965).

Figure 4 shows a comparison of the ice thickness transect inferred from our gravity survey with nearby values determined by Pelto et al. (2020). Our profile is mostly within 100 m of the radar-based measurement locations, but there is only direct overlap at the ends. Most values are in broad agreement with nearby radar measurements, with minor discrepancies at both profile ends. At the western end, the gravity base station was located on exposed bedrock, co-located with a radar-based ice thickness measurement in the 1-20 m bin, implying true loss of ice at the glacier edge between 2017 and 2023. At the eastern end of our profile, the gravity-derived value of $51 \pm 2$ m is lower than the radar value that is binned into the 60-80 m range. While this could also be due to actual changes in the ice thickness over the 6 years between surveys, it may be explained by the gravity-derived values reflecting averaged ice thickness over a larger area, in contrast to the more direct radar point measurements. Areas of lower ice thickness (40-60 m range) were measured by Pelto et al. (2020) approximately 15 m to the east of the end of our profile and ∼55 m to the northwest and northeast (Fig. 4). Accepting the few differences, the similar values and trend derived from two independent methods applied to Kokanee Glacier verify that a gravity survey can enable a good first-order approximation of glacial ice thickness, and provide confidence in our results for Comox Glacier, despite a lack of groundtruthing data there.

# 6 Conclusions

Collection and modelling of gravity survey data along transects across Comox Glacier and Kokanee Glacier has enabled determination of average ice thicknesses of $42 \pm 4$ m and $50 \pm 3$ m, respectively. This study provides the first ice thickness measurement for Comox Glacier, to serve as a baseline to which future measurements can be compared, and to inform decisions around water resource management in the Comox Valley watershed. The ice thickness profile of Kokanee Glacier was found to be consistent with previous measurements obtained using ice penetrating radar, and the thickness of both glaciers was found to be comparable to other systems of similar surface area.

Gravity surveying, while not a commonly applied technique in recent glacier studies, is demonstrated here to be a viable and cost-effective method for the estimation of glacial ice thickness to within 5-10%. If applied more widely, it has the potential to significantly add to the global ice thickness database. Such data are critically needed to constrain regional and global assessments of glacial ice volumes and their changes over time, with implications for water supply, glacioisostatic adjustment, and relative sea level change (e.g., Farinotti et al., 2017; Raper and Braithwaite, 2005; Shugar et al., 2014).



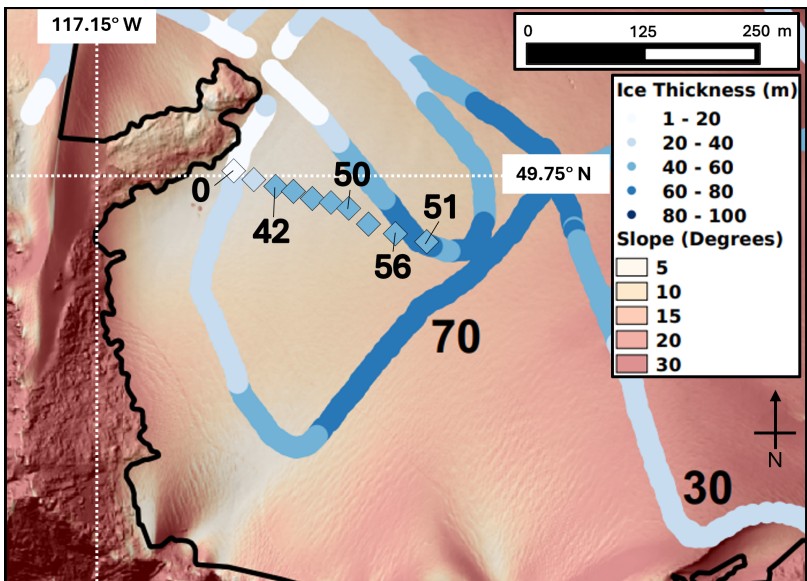

**Figure 4.** Comparison of ice penetrating radar measurements of ice thickness (broad lines) and gravity-derived ice thickness values from this study (diamonds) for the same region of Kokanee Glacier. Both data types are coloured by ice thickness ranges; some specific values are also provided. Base figure modified from Pelto et al. (2020).

**Supplement:** The supplement related to this document is available at: https:/doi.org/xxxx

195 **Author contributions:** Conceptualization: M.R.G.F, L.J.L. Formal analysis: M.G.R.F. Investigation: M.G.R.F. Resources: L.J.L. Supervision: L.J.L. Validation: L.J.L, M.G.R.F. Visualization: M.G.R.F. Writing - original draft: M.R.G.F. Writing - review and editing: L.J.L, M.G.R.F.

**Competing interests:** The authors declare that there were no known competing financial interests or other relationships that could have influenced the findings of this research.

200 **Acknowledgements:** The authors would like to thank Josh Forbes and Aileen Blakeman for their unquestioned assistance with transport, mountain traversal, and the relative elevation survey during data collection. Samantha Palmer is also thanked for providing a MATLAB version of the solid Earth tide program by Ahern (1993).

**Funding:** This work was supported by the Summer Undergraduate Research Award (SURA) to M.R.G.F. from the Faculty of Science at the University of Victoria.



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
