# Peer review of "Ice thickness of Comox and Kokanee Glaciers, British Columbia, determined through relative gravity surveys"

_EGUsphere, 2025_

## Author Comment (AC1)

**Response to Reviewer Comment #1 (**https://doi.org/10.5194/egusphere-2025-3456-RC1**) – please see our response in blue, beneath each review comment.**

This manuscript presents an interesting and clearly written study applying gravimetric methods to estimate glacier ice thickness. The approach is generally well explained, and the results appear reasonable. The paper is concise and direct, which is appreciated. However, I find that the manuscript is currently too limited in scientific depth to warrant publication in its present form. With targeted additions, it has the potential to become a valuable contribution.

*We thank the reviewer for their comments and suggestions, addressed below.*

To increase scientific relevance, I recommend strengthening the manuscript by expanding the analysis beyond the current two-site application. Options could include adding a comparison with existing model-derived ice thickness estimates (e.g., those from Farinotti et al.) to contextualize their results. Given that Comox Glacier (e.g., GLIMS ID G234649E49550N; RGI ID RGI2000-v7.0-G-02-07858) is likely included in global modelling datasets, this comparison could help quantify the method's performance and assess any bias. For example, if the measured thickness is much lower than estimated by other modelling approaches, what are the implications for retreat and disappearance of this glacier (which can also be obtained from global estimates such as Rounce et al., 2023)?

*This is an excellent suggestion – we have now checked and both glaciers are indeed included in the Farinotti et al. database. In our revision, we will add a comparison of our gravity-derived ice thicknesses with these model-derived estimates, and will consider the implications for glacial retreat, and within a broader context, as suggested.*

Additionally, incorporating a discussion of methodological uncertainty (is the varying density of crevasses/melt pockets likely to cause a big shift in the results? Is the grid size a big influence?), including potential sources of divergence between gravimetric and radar measurements and whether measurement year might influence results, would significantly enhance the scientific contribution. These are just initial thoughts on how this manuscript would be expanded, but I am certain there are other avenues as well.

*This is also a good suggestion. As alluded to in the paper, the presence of crevasses and melt pockets or gravel inclusions within the ice will have some influence, but this is expected to be minor and competing effects will at least partially cancel each other out. We don't expect much change in the results, but we will carry out some quantitative testing based on the literature (e.g., mean 0.8% water content estimated by Pettersson et al., 2004). As for the influence of the topographic elevation grid used for calculation of the terrain correction, only very minor differences would be observed in the final solution under a finer grid size than 25 m. With only small-scale deviations (i.e. generally less than a meter) taking place in the topology over short distances around each station, it is expected that taking a finer grid size would introduce a negligible change to the topological corrections (generally a smaller magnitude gravity correction here) and thus modelled ice thickness, while simultaneously increasing computational demand. Therefore, while it may be important for other studies under more varied short distance terrain, in this study we believe a 25 m grid to be sufficient.*

**Minor comments**

Please include the GLIMS and/or RGI glacier identifiers in the study site description to facilitate spatial reference.

*We will add the identifiers in the revised version.*

In the introduction, the Scott citation focuses on the Hindu Kush region only; consider incorporating a broader reference such as Viviroli et al. (2020) for global or multi-mountain context.

*We thank the reviewer for this useful suggestion and reference, which we will cite in the revised version.*

Figure 4: clarify the distinction between your measurements and those from Pelto (2020) in the figure itself, not just the caption.

*We will make this change to improve clarity in the revised figure, along with several others suggested by the other reviewer.*

References:

Viviroli, D., Kummu, M., Meybeck, M., Kallio, M. & Wada, Y. (2020). Increasing dependence of lowland populations on mountain water resources. Nature Sustainability, 3(11), 917–928. https://doi.org/10.1038/s41893-020-0559-9

Rounce, D. R., et al. (2023). Global glacier change in the 21st century: Every increase in temperature matters. **Science**, 379(6627), 78-83.

Pettersson, R., Jansson, P., & Blatter, H. (2004). Spatial variability in water content at the cold-temperate transition surface of the polythermal Storglaciären, Sweden. Journal of Geophysical Research: Earth Surface, 109(F2), https://doi-org.ezproxy.library.uvic.ca/10.1029/2003JF000110

---

## Author Comment (AC2)

**Response to Reviewer Comment #2 (**https://doi.org/10.5194/egusphere-2025-3456-RC2**) – please see our response in blue, beneath each review comment.**

**General Comments**
This manuscript provides a concise presentation of a method for estimating glacier ice thickness using gravity survey data and an inversion modelling approach. The rationale for the study is sound: ice thickness data are sparse, yet glacier thickness information is valuable for projecting ice volume losses. I recognize the work that went into this manuscript, and I recognize the value it provides to the community, however, in the current form I think this manuscript has a couple of deficiencies that should be addressed before publication:

1. Validation: There is simply not enough validation data presented to demonstrate the accuracy of this approach. There needs to be a more rigorous comparison between ice thickness estimates derived from the gravity survey method, versus ice thickness measured or modelled using other well established methods. The comparison that was done here using survey data from Kokanee Glacier was a good attempt, but unfortunately, a bit of a miss. I'll go into more detail on this below, but the fact is, there are only 2 (maybe 3?) points of overlap between gravity survey points presented here and the prior ice penetrating radar survey by Pelto et al. (2020). The fact that the average ice thickness estimated here, compares well with the average ice thickness determined by the Pelto et al. (2020) study is a good indication, but average ice thickness is only so useful.

2. Methods: This manuscript presents a "novel method" (old technology, revived and updated for this application), and the authors have a good grasp on the underlying physics, the survey methods, and the calculations required to perform the inversion modelling. However, I'm not confident I could reproduce the results, given the methods presented here. Again, I'll go into more detail and provide examples below. But if the goal of this paper is to present "a lower cost and logistically simpler alternative" method to measuring ice thickness, then I think some more work needs to be done to sell this alternative to glaciologists who are already very comfortable and confident with ice penetrating radar. I encourage you to proceed, because I think there is value in developing methods that could indeed be lower cost and easily field transportable, but I think you could make the value more apparent, and the method more approachable.

*With these two points outlined in detail by the reviewer below, we will similarly address them in more detail below.*

**Specific Comments**
**Figures:**
Figure 2 could use some work. Addition of contour lines on glacier maps would aid interpretation, or perhaps using color images rather than grey-scale would make the orientation of the glaciers clearer.

*We will work to improve the clarity of Figure 2, with the addition of topographic contour lines and/or colour imagery.*

Photos or diagram of instrumentation setup/operation would aid communication of core methods

*We don't think this is necessary within the main paper, since methods to measure relative gravity and elevation are both long established and partially dependent on instrument model. We can certainly include a figure in the supplementary material, along with a more detailed description of the method, as requested by this reviewer.*

Figure 3: You could take this one step further. The second panel in each sub figure is your modeled ice depth. If I understand correctly, the polygons are ... a way of triangulating the depth to the bottom of the gravity anomaly (ie. ice). This feels like a step in the process of getting towards an ice profile. Could you make one additional panel where you plot the surface elevation and bed elevation of the ice, vs distance long transect?

*This is a good suggestion and we are considering it, though such a profile looks very similar to the existing second panel (with a minor amount of downward surface tilt with distance from each base station), especially for the Kokanee transect, which has a maximum surface elevation change of only 3 m, compared to ice thickness of up to 56 m. We will remove the polygon boundaries in the revised figure (2nd panels), since they are artificial (adjoining polygons is a simple way to construct a single ice body in the modeling process), but perhaps misleading.*

Figure 4 should really be one of the primary figures you highlight. At the very least, move this figure up into the Discussion section, so that it can be referenced alongside your discussion of this comparison dataset.

*We agree this is a key figure and we will move it up to appear within the Discussion section where it is described.*

The remainder of this is organized based on my 2 main criticisms above: 1. Validation, and 2. Methods

1.    **Validation**

The ice penetrating radar (IRR) measurements from Pelto et al. (2020) are the most direct data for comparison to and validation of results presented in this study.  The biggest question that comes to my mind is: why didn't you follow one of the IPR transects done by Pelto et al. when the field survey was conducted on the Kokanee Glacier?  Only 2 (maybe 3) of your measurement points overlap with the IPR measurement transects.  If you had directly overlapped one or more of those prior transects, you would have had a very nice comparison dataset that could be used to make a robust case for your method.  Instead, a reader is left to guess why this transect was chosen for the gravity survey.  I'm sorry to say, but this comes across as the main design flaw in this study.

*The transect location was chosen for the Kokanee Glacier over what we believed was the safest route for a ~500 m transect that would also meet the scientific objective of crossing the main glacier trunk. While it is true that the transect chosen only directly overlaps a few locations of the Pelto et al. (2020) IPR measurements, we don't believe this should be a major concern. Unlike drilling a borehole which would provide the actual ground-truth ice thickness at a single point, a gravity measurement is closer to determining the average ice depth over a larger area. The ice thickness retrieved at a specific station in this study, while presented as a point, should instead be thought of as an average around that point which becomes less accurate with distance. While we agree with the reviewer that directly following a transect by Pelto et al. (2020) would have made for an easier comparison, we believe that the close distance at the start and end of our transect to that of Pelto et al. (2020) is sufficient to look at how the gravity survey compares to current methods.*

A further question about the IPR data: was there communication with the authors of the Pelto et al. study to ask for access to their ice thickness data?  The reason I ask is that I don't understand why the IPR ice thickness data is binned into 20m thickness intervals.  That makes it even more difficult to compare your measurements with theirs, at a meaningful level of precision.

*This is a good suggestion – no, we were not previously in communication, but we now plan to contact the Pelto et al. authors to request access to their higher-resolution data, to enable a clearer comparison with our gravity-derived ice thickness values.*

Line 171-172: *"At the eastern end of our profile, the gravity-derived value of 51 ± 2 m is lower than the radar value that is binned into the 60-80 m range."*

This is the one location where you have overlapping measurements to directly compare ice thickness values, but it feels very unsatisfying to compare your measurement (at ~2m level of precision) to a value in a 20m range.

Even if there are a small number of direct cross-over points between the gravity survey measurements and the IPR transects, I think you have another option to explore.  The authors

of the Pelto et al. paper went on to generate a gridded model of ice thickness for the entire Kokanee Glacier. I suggest you extract a 2D profile of ice thickness from their gridded model of Kokanee Glacier, and compare this to the modelled ice thickness profile based on the gravity survey (ie. Figure 3).

*This is also a good suggestion. The published gridded model for Kokanee Glacier (in Fig. 5 of Pelto et al. 2020) has too low of a resolution to extract useful values – we plan to request access to the model data in order to facilitate such a comparison.*

Related:
Line 171*:"... thickness measurement in the 1-20 m bin, implying true loss of ice at the glacier edge between 2017 and 2023."*

Line 173*:" While this could also be due to actual changes in the ice thickness over the 6 years between surveys,..."*

Pelto et al. (2020) published observed and modelled mass balance gradients for all of the glaciers in their study. You could back up your statements in the lines noted above by calculating the expected ice thickness loss from 2017 – 2023 using the mass balance rate for the elevation of your gravity survey transect. This would be better than speculating.

*We thank the reviewer for the suggestion – we will make the calculation and revise the text accordingly.*

2. **Methods**

I appreciate the presentation of an alternative / complimentary method to ice penetrating radar for ice thickness determination. In my opinion, your goals for this manuscript should be: 1) present the gravity survey method for collecting measurements that yield ice thickness data, 2) demonstrate that the results approximate those achievable from IPR, 3) clearly outline the requirements and procedures for conducting the field survey and data post-processing so that glaciologists can assess whether this is a practical alternative to IPR methods, 4) discuss pros and cons compared to other methods of determining ice thickness. And, if you really want to make the argument that the gravity survey method is "a lower cost and logistically simpler alternative" then you need to demonstrate this.

Line 42-44*: "Gravity equipment is more portable than other in situ instruments, and surveys can be completed relatively quickly with a team of just two people, suggesting that gravity surveying is an overlooked but valuable method of determining glacial ice thickness."*

This is the opening statement of this argument, but then not followed up with further detail. More portable? Survey's are relatively quick? This sounds nice, and I'd like to know more. What does your total field kit look like? How much does it weight? How long does it take to make a measurement? I read later on that it takes 2 hours to make 12 measurements.

*Please see our response to these linked comments below.*

Line 39-42 *"A drawback of the gravity method in the past was the need for tedious and time-consuming terrain corrections … such corrections are now much simpler and quicker, using digital elevation models and standard computer routines (e.g., Cella, 2015)."*

This seems to be the main innovation that you are bringing forward. Perhaps this needs to be explained in the methods section, so that prospective users can evaluate the practicality of implementing this method in the field.

*Please see our response to these linked comments below.*

Section 3.1 Gravity field surveys
The questions When and Where have been answered. What you did is briefly described. What you used (equipment) is introduced, but some of the equipment seems to be interchangeable or perhaps optional (Total Station)? How it all works together is what I feel is missing.

Visuals would help. Assume I don't know what a gravimeter looks like, or how it works. Show me the whole field setup, so I understand what I would need to carry and how I would set it up. A diagram that demonstrates the field measurement and how it relates to the gravity anomaly and the influencing factors would be very valuable.

*For our revised manuscript, we will add a section to the supplement about what a researcher would need, what the essential field kit would look like, how much it weighs, and more about the practical methods; however, we feel that with gravity surveying being an established and well-documented method outside glaciology, an in-depth discussion around the theory, corrections, and inner workings of a gravity meter is unnecessary. While we agree a greater discussion about the logistics may be helpful, we believe it is a discussion better suited to the supplementary material, leaving the main text to focus on the results of the experiment, and to ask if gravity surveying is an effective scientific method for gathering ice thickness data. For those researchers who agree with the efficacy of this method in the context of glacier studies, we will direct them to our supplementary for the logistical details, and towards well established literature for the fundamentals of gravity surveying.*

How and Why did you choose the measurement transects on these glaciers?

*The transects were chosen as the safest ~500 m straight line paths which could both (1) meet the scientific objective of crossing the main trunk of the glacier and (2) be traversed in a single data collection session without needing extensive mountaineering equipment.*

Line 83-85: If you had a GPS with you, and used this to record the location and elevation of each gravity survey position, why did you need the Total Station? Unless you need this for some specific reason, it seems unnecessarily cumbersome to carry and time consuming to setup and use at each measurement point. If a Total Station is required for the gravity survey,

please explain why, or explain how GPS could be used as an acceptable alternative (or not, as the case may be).

*The handheld GPS was used to record the approximate absolute position (latitude, longitude) of each measurement position. However, a handheld GPS is insufficiently accurate to determine horizontal and vertical distances between points. These distances must be determined accurately (within a few centimetres) to enable corrections to be made to the gravity data.*

Line 92-92: On the Kokanee Glacier, you used a laser rangefinder to measure relative height and distance along transect, instead of the Total Station. Again, why was this necessary if you had a GPS? Is there a level of precision you require in your along-track distance that you could not get via GPS coordinates? If so, please explain.

*Indeed, the various gravity corrections require relative elevation to be known within a precision of a few centimeters. A handheld GPS has typical elevation uncertainty of at least several meters, which is not sufficient.*

And when using the Bosch laser rangefinder, how did you measure relative height? I'm guessing the rangefinder has an inclinometer, so you could measure the angular distance and then derive the height change between points. What were you using as measurement targets; were these targets consistent; did you have to move the targets from point to point?

*The rangefinder model we used does indeed have a built-in inclinometer, such that it provides both the horizontal distance and the derived vertical distance between two points. The measurement target was a rigid plastic flag held at a consistent height above ground, moved from point to point by a second person.*

Section 3.2 'Data reduction & density modelling'
This section is well written as a brief description of what was done, but as someone without prior familiarity with gravity survey methods, I'm left with a lot of questions. For instance, what is a Bouguer anomaly?

*We will add more clarification to the text with these considerations in mind. A Bouguer gravity anomaly is the difference in observed gravity that results only from differences in the density of subsurface material. To arrive at the Bouguer anomaly means making corrections for all other factors that affect gravity observations (differences in latitude, elevation etc).*

I feel that perhaps the Supplement section has been relied on too heavily.

While I don't expect a step by step manual, I would like to see clearly presented: an overall explanation of the method, a list of the measurements and parameters required, and a list (or table) of the parameters used in the study to derive the results.

*With gravity surveying being an already established modern method outside glaciology, we feel including an in depth explanation of the practical methods is best suited to the supplementary material. The major focus of this study was to ask if gravity surveys could be used to derive ice thickness with accuracy comparable to current methods, and to provide new data. While we feel it is important to note some logistical benefits, we believe it would unnecessarily lengthen the main text to provide more in-depth theory there. As such, we will strengthen the discussion about the practical methods in the main text, but would prefer to leave the details of the many gravity corrections and their theoretical basis to the supplementary material.*

As just one example, I really don't understand the terrain corrections or the Bouguer plate corrections. How did you take the necessary measurements to make these corrections? In the field, standing at each measurement point, it seems you need to imagine concentric circles around your position, and measure distances to points... Are these data presented in Table S2? How sensitive are your results to these corrections? This is part of the overall method that seems very obscure to me, even with the description in Supplement S2.4. This is the reason for my earlier comment that I would not be confident that I could reproduce your results.

*Indeed, the results are somewhat sensitive to the corrections discussed. All information needed to make the corrections is either provided in Table S1, or provided by GeoBC (2013) topographic maps, which are used by the software to extract data from concentric rings. In the revised version of the manuscript we will work to include more and clearer information surrounding the calculation of each gravity correction; however, with gravity surveys being such a well-established modern method outside glaciology, we believe it would not add much value to go into great detail. The detailed theory can already be found in many texts (e.g., Lowrie and Fichtner, 2020), and we feel directing the reader to these sources is more beneficial than repeating it here.*

*Lowrie W, Fichtner A. Gravity and the Figure of the Earth. In: Fundamentals of Geophysics. Cambridge University Press; 2020:48-86. https://doi.org/10.1017/9781108685917*